# Association between Breakfast Meal Categories and Timing of Physical Activity of Japanese Workers

**DOI:** 10.3390/foods11172609

**Published:** 2022-08-28

**Authors:** Farnaz Roshanmehr, Katsuki Hayashi, Yu Tahara, Takahiko Suiko, Yuki Nagamori, Takao Iwai, Shigenobu Shibata

**Affiliations:** 1Laboratory of Physiology and Pharmacology, School of Advanced Science and Engineering, Waseda University, Shinjuku-ku, Tokyo 162-8480, Japan; 2Research and Development Headquarters, Lion Corporation, Edogawa, Tokyo 132-0035, Japan

**Keywords:** chrono-nutrition, circadian clock, sleep, breakfast, exercise

## Abstract

Background: Breakfast is the most important meal of the day and has been associated with longevity. Regular breakfast consumers often have a healthy lifestyle, including a healthy diet and regular physical activity. Methods: We examined the association between breakfast type, chronotype (morningness-eveningness), and physical activity in 3395 Japanese workers using a cross-sectional web survey. Results: Participants who ate Japanese breakfasts showed an early chronotype, while those who ate breakfast cereal exhibited a later chronotype. Physical activity was positively associated with adopting a Japanese breakfast style. Japanese breakfast eaters performed physical activities from 6:00–9:00 compared with other breakfast eaters. Conclusion: Our findings suggest that eating a Japanese breakfast is associated with an earlier chronotype (morningness) and higher physical activity.

## 1. Introduction

Breakfast is generally defined as the first meal of the day eaten within two hours of waking up [1]. Many studies have indicated the association between breakfast consumption and a lower BMI [2,3], followed by a decrease in the prevalence of obesity-related chronic diseases [1], including coronary heart disease (CHD) [4] and type-2 diabetes [5]. On the other hand, the limited evidence of breakfast consumption as an important factor in combined weight and cardiometabolic risk management is suggestive of a minimal impact [6]. Regular breakfast consumers often have a healthier lifestyle, including healthy dietary intake [7], lower snacking habits [8], and higher physical activity [9]. Breakfast skipping may lead to a decrease in free-living physical activity during the day [10] and may also decrease exercise performance [11]. Furthermore, obesity and overweight are also associated with skipping breakfast [12,13,14]. These papers suggest that eating breakfast is particularly relevant for individuals who want to maximize their exercise performance.

Breakfast skipping habits may differ among breakfast categories. For instance, there is a growing trend in Western society of skipping breakfast, with a report indicating that 36% of the UK population tends to skip breakfast almost always [15]. In addition, there is a correlation between increased physical activity and fewer health problems in boys and girls [16]. Furthermore, poor quality of life is associated with a child’s unhealthy lifestyle, including skipping breakfast, lack of physical activity, and late bedtime [17,18].

Morningness-eveningness, called the “Chronotype”, is associated with the circadian phase [19]. Attention and cognitive functions show synchrony with chronotype; morning-type individuals perform better in the morning, with worse execution in the evening, while evening-type individuals show an inverse pattern [20]. Evening-type individuals’ biological circadian rhythms, such as cortisol levels, body temperature, and the sleep-wake cycle, are delayed compared with morning-type individuals [21,22,23]. Habitual physical activity in young adults is positively associated with them being morning-type individuals [24]. Physical activity has been linked to improved self-esteem and reduced anxiety, according to a research review on the relationship between physical activity and mental health among young people [25]. However, it is crucial to monitor whether physical activity and its timing, along with eating breakfast, would affect and improve the health of Japanese workers. Studies have shown that managing employee health at work can improve employee performance and the company’s performance. This issue is becoming more prevalent globally [26,27,28].

However, these studies are still insufficient to provide strong evidence regarding the relationship between different breakfast types, chronotypes, and physical activity. Therefore, in this study, we aimed to investigate whether there is an association between breakfast meal categories and the timing of physical activity.

## 2. Materials and Methods

### 2.1. Ethical Approval

The Ethics Review Committee on Research with Human Subjects at Waseda University and Lion Corporation approved this experiment (No. 2020-046 and No. 349, respectively), and the guidelines laid down in the Declaration of Helsinki were followed.

### 2.2. Target Population and Data Collection

In this cross-sectional study, we screened 5534 participants who responded to a questionnaire. We used the same samples published in our previous paper [28]. Participants were recruited through an online survey company (Cross Marketing Inc., Tokyo, Japan) from 19 December 2020 to 25 December 2020. Of the 5534 participants, 808 and 1331 participants who were not working during the daytime and those who did not answer the International Physical Activity Questionnaire (IPAQ), respectively, were excluded. Therefore, a total of 3395 participants (2352 men and 1043 women) were selected for the study. All the participants were Japanese workers aged 20–69 years living in Japan. According to a Japanese national survey from the Statistics Bureau, the sex ratio of the participants in this study was like that of full-time Japanese workers (https://www.stat.go.jp/english/data/index.html; accessed on 1 June 2021). Informed consent was obtained from all the participants.

### 2.3. Variables

According to the type of work (shift work or other), there were 288 to 333 questions, the number of questions depended on the participant’s working condition (e.g., 288–307 for normal work, 304–323 for shift work, or 314–333 for remote homework) and participants were given 20–30 min to answer them. Thus, in this study, we focused on the aforementioned variables. The basic participant characteristics included age, sex, body mass index (BMI), ingredients consumed in breakfast meals, and timing of eating breakfast. To obtain sleep parameters, participants were asked about their sleep onset and wake-up times on weekdays and weekends (free days), and the sleep length was calculated. Using the Munich Chronotype Questionnaire (MCTQ), chronotypes were assessed from the mid-point of the sleep phase (from onset time to wake-up time) on free days. There are validation studies of the MCTQ in Japanese participants, and the Japanese version of the questionnaire is widely used [29].

Breakfast type participants were asked which of the following five categories is their preference: Japanese breakfast (J), Western breakfast (W), a combination of Japanese breakfast and Western breakfast (J-W), cereals (C), and skipping breakfast (SB). Data were analyzed as 0 (no) or 1 (yes) for each breakfast type. In the same question, we explained that J includes rice as the main dish and W includes bread as the main dish. The details of each breakfast style are as follows. J includes rice and very often miso soup; W includes bread and salad; and C includes cereal and milk or soymilk. In Japan, it has already been demonstrated that one Japanese pattern style is rice/vegetable/fish/pulse/seasoning, while the Western style is bread/dairy/fruit/sugar [30].

Daily physical activity was objectively measured using moderate-to-vigorous physical activity (MVPA) [31]. However, in the present experiments, a short version of the self-reported International Physical Activity Questionnaire (IPAQ) was used to evaluate daily physical activity [32]. Hence, the participants were asked about the number of days and hours spent doing each of the three types of activity during the last seven days: vigorous-intensity activity, moderate-intensity activity, and walking. We then calculated weekly metabolic equivalents (MET) for the following four activity intensities based on the IPAQ analysis guidelines [33] and used the following formulas: walking MET (min/week) = 3.3 × walking min × walking days; moderate MET (min/week) = 4.0 × moderate-intensity activity min × moderate days; vigorous MET (min/week) = 8.0 × vigorous-intensity activity min × vigorous-intensity days; total physical activity MET (min/week) = sum of walking + moderate + vigorous MET. According to the Ministry of Health, Labor and Welfare in Japan, the standard for physical activity for individuals aged 18–64 years is 23 MET-h/week, that is, 1380 MET-min/week of physical activity with an intensity of ≥3 METs [34]. In the next question, we asked about the timing of each vigorous, moderate, and walking physical activity from (1) 00:00–03:00, (2) 03:00–06:00, (3) 06:00–09:00, (4) 09:00–12:00, (5) 12:00–15:00, (6) 15:00–18:00, (7) 18:00–21:00, and (8) 21:00–00:00.

Short version IPAQ is validated in Japanese participants, and now the Japanese version of the questionnaire is widely used [35,36].

### 2.4. Statistical Analysis

Statistical analyses were conducted using the SPSS software (version 27, IBM, New York, NY, USA). The data were interpreted as the mean ± standard deviation (SD) or the mean ± standard error (SE). Statistical significance was set at *p* < 0.05. The statistical significance of the two groups was determined by Mann–Whitney analysis, except for the breakfast style, which was analyzed by Fisher’s exact probability test. Spearman’s correlation analysis was performed between basic characteristics, exercise, sleep parameters, and breakfast meal style. For data with three or more groups, normal distribution and equal variations were examined using the D’Agostino Pearson/Kolmogorov–Smirnov and Bartlett’s tests, respectively. Non-parametric analysis was conducted using Kruskal–Wallis and Dunn’s post-hoc tests. Moreover, multivariate linear regression analysis for the interaction between various exercise factors and breakfast style was conducted.

## 3. Results

The mean age of the participants was 49.7 and 44.8 years for men and women, respectively. The participant characteristics are presented in Table 1. The differences in age, BMI, strong and moderate physical activity, eating breakfast cereals, and sleep parameters were found to be significant among both sexes, whereas walking activity, eating Japanese, Japanese/Western, Western, and skipping breakfast did not differ significantly. In the current research, most participants tended to eat W more than J.

The correlations between basic characteristics, exercise, sleep parameters, and breakfast meal style were analyzed (Table 2). As indicated in Table 2, there was a positive association between BMI and eating a Japanese and J-W breakfast. The most important point indicated in this table is that there was a positive association between strong/moderate physical activity and the J type. However, it showed a negative association with the W type. Additionally, there were positive associations between sleep factors (wake-up, sleep onset, and MSFsc) and skipping breakfast. Sleep factors showed negative associations with eating J (Table 2), suggesting that participants who eat J are morningness individuals.

Next, the interaction between age, MSFsc, and physical activity was assessed (Figure 1). In both sexes, aging people tended to eat W (Figure 1A). In addition, the participants who used to eat J had an earlier chronotype than those in the other groups; meanwhile, participants who skipped breakfast meals showed a later chronotype (Figure 1B, Table 2). Furthermore, J eaters showed an association with strong and moderate activity. Male J eaters had higher METs in strong and moderate activities (Table 2, Figure 1C).

Furthermore, a multivariate linear regression analysis was conducted to examine the interaction between physical activity, sleep, and breakfast style (Table 3). In accordance with what is shown in Figure 1, this study also demonstrates the positive interaction between strong/moderate activities and J. Table 3 also demonstrates that there was no association with other types of breakfast in cases of vigorous, moderate, and walking activities. There is a strong negative association between MSFsc and J breakfast (Table 3).

In order to check further the relationship between physical activity and breakfast style, binomial logistic regression was conducted (Table 4). We divided the objective data into two groups following its variation (smaller than the median as “0” or more than the median as “1”). However, more than half of METs data for strong exercise and moderate exercise were 0 (strong: *n* = 2157, moderate: *n* = 1779). Thus, we split these data in the following order (data = 0:”0”, data > 0: “1”). J and J-W styles were significantly associated with physical activity or moderate exercise. Eating C and walking exercise were also significantly associated. However, the odds ratio regarding the Japanese style (J and J-W) tended to be bigger than the others. Thus, Japanese-style breakfast might have more association with physical activity than other styles.

Figure 2 shows more details about the timing of physical activity and eating breakfast types. As shown, at different time points, almost all J eaters had strong, moderate, and even walking activities at an earlier time (3–6, 6–9 o’clock) than other breakfast groups. Moreover, as shown in Table 3, physical activity was well associated with the J style. Therefore, here we compared the J group and other groups with a daily time course (Figure 2B). This figure shows that J eaters engaged in strong and moderate activities earlier in the morning than other breakfast groups, while walking showed a similar pattern.

## 4. Discussion

This was a cross-sectional study with a large number of Japanese workers (*n* = 3395) aged 20–69 years. We revealed, for the first time, an association between different breakfast meal categories and the timing of physical activity. In a study by Thapa et al., a positive association between evening chronotype and daily physical inactivity was found. This paper suggests that morning chronotype may be associated with high physical activity. Our results demonstrate that participants who skipped breakfast meals had a later chronotype, which is in line with the results of previous studies [37,38]. Thus, skipping breakfast may lower physical activity because of the evening chronotype. The present results demonstrate that the difference in MSFsc was 0.7 (male) and 1.1 h (female) between the J and skip groups. Weekend sleep and wake times were 23.5 and 7.2 for the J group and 24.4 and 8.2 for the skip group in females. Thus, almost 1 h difference in sleep or wake times appeared. At present, we do not know whether these differences are involved in real-life scenarios or not. Follow-up of 2.1 years delayed 40 min of MSFsc in adolescence, and delta MSFsc was positively associated with Fat-Free Mass index [39]. Thus, 0.7–1.1 h differences in MSFsc in the present research may be related to muscle function.

In our study, relatively more participants tended to eat W than J. However, there was a negative association between strong and moderate physical activity and eating W type. Moreover, the results showed that older people ate W more than other breakfast types. Traditionally, WS contains high carbohydrate content, and research has demonstrated that skipping breakfast affects dietary profiles, particularly because carbohydrate consumption is reduced [40,41]. In addition, the consumption of breakfast has favorable effects on carbohydrate metabolism parameters compared with that of skipping breakfast [42].

Moreover, many studies have shown that morning carbohydrate consumption can easily enhance exercise performance compared with skipping breakfast [43,44]. In addition, the outcome of research by Chryssanthopoulos et al. showed that consuming a high carbohydrate breakfast meal 3 h before physical activity can improve the endurance running capacity by approximately 9% compared with skipping breakfast [45]. However, participants were all male and trained runners—albeit amateur. Therefore, a 9% increase in endurance capacity is not expected in the current study population because strong and moderate METs did not show big differences between J and skipping breakfast groups.

More studies indicate that skipping breakfast can reduce exercise performance due to the limitation of glycogen in muscle metabolism [11,46]. Hence, this evidence suggests that for optimal exercise performance, breakfast should be consumed between 1 and 4 h before exercise. However, Maraki et al. suggested that evening exercise may be more tolerable than morning exercise [47]. To focus on the importance of this issue and skipping breakfast, a study performed a trial and found that a 4–5 % decrement in exercise performance was observed following breakfast omission [48]. Moreover, these outcomes suggest that breakfast has long-lasting effects; therefore, consuming it is important for individuals who want to maximize their exercise performance.

J consumption is associated with morningness, and people with morningness showed higher physical activity than those with eveningness. In the current study, J eaters showed high METs for strong and moderate physical activity. This result strongly suggests that habitual intake of J might be beneficial for health because of its morningness and high physical activity. Interestingly, a Japanese survey paper demonstrating that a rice-based breakfast pattern is a healthier diet than a bread-based breakfast pattern [49] supports the above possibility. The results of the present study replicate and clearly strengthen the findings of previous studies that suggest that the Japanese Food Guide should be promoted to establish a healthy eating pattern and prevent obesity among the Japanese population [50]. A study conducted in Japan also reported that eating breakfast daily plays a crucial role in preventing obesity [51]. However, the current study shows a strong positive association between BMI and eating Japanese and J-W breakfast meals. The participants eating Japanese or J-W breakfasts were younger than those eating W, while younger individuals showed higher BMI because of larger muscle volume. In addition, younger individuals tend to have a greater Fat Mass to Fat-Free Mass (FFM:FM) ratio than older individuals as a consequence of age muscle atrophy.

The percentage of people with strong and moderate physical activity at approximately 6–9 o’clock was significantly higher in the J group than in the other groups (Figure 2B). A study by Miyazaki et al. showed that physical wellness contributed to the morning and evening physical activity preference. In addition, they showed that physical activity preference is associated with physical wellness in the morning [52]. Moreover, it has been reported that exercise in the morning to early afternoon advanced the circadian rhythm, whereas exercise in the late evening delayed it [53]. This study, along with this result, may support the idea that J helps to maintain morningness through a physical activity during the morning.

The present study has several limitations. For instance, there was an unbalanced sex ratio (approximately 70% men). In addition, data collection was self-reported and not validated. We performed this study during the global COVID-19 pandemic (not lockdown), which may have affected the lifestyles of Japanese workers [54,55].

Furthermore, our study design did not imply causality. Nevertheless, our findings suggest that eating J may be a predictor of an earlier chronotype (morningness) and having higher METs, which is the ability to engage in vigorous and moderate activities. Interestingly, Morningness-Eveningness Exercise Preference Questionnaire (MEEPQ) in Japanese university students certainly suggests that there is a relationship between MEEPQ scores and objectively measured physical activity [52]. In addition, they showed that physical activity preference is associated with physical wellness in the morning [52]. This paper suggests that MCTQ and/or MEEPQ are good questionnaires to predict daily physical activity [52].

Hence, long-term intervention studies are required to confirm this relationship and to determine the impact of eating J on the timing of physical activity.

## Figures and Tables

**Figure 1 foods-11-02609-f001:**
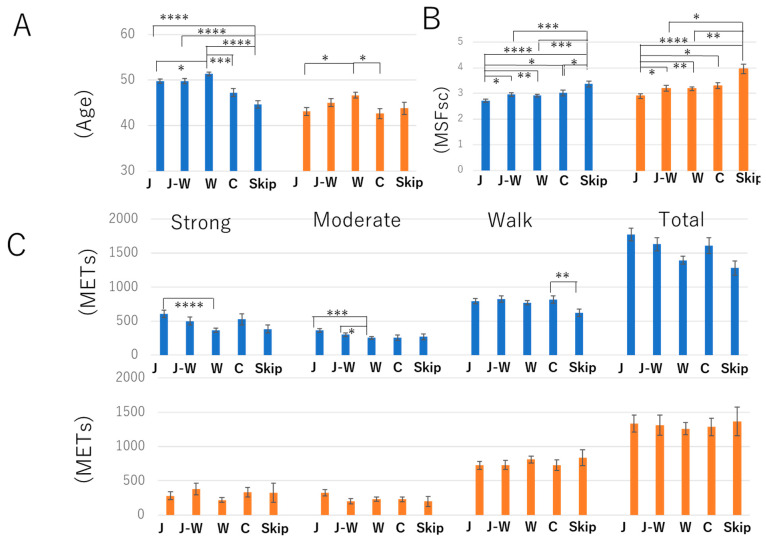
Interaction between various parameters and breakfast meal category. Preferences for each meal category. Association between age (**A**), MSFsc (**B**), exercise (strong, moderate, walk, and total) (**C**), and meal category. The blue and orange columns indicate men and women, respectively. Statistical significance was determined using the Kruskal-Wallis test followed by Dunn’s multiple comparison post-hoc test: * *p* < 0.05, ** *p* < 0.01, *** *p* < 0.001, **** *p* < 0.0001.

**Figure 2 foods-11-02609-f002:**
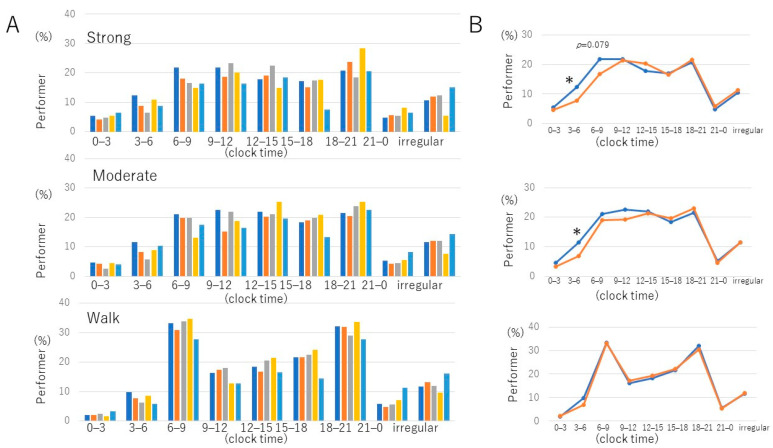
Performer percentage to each exercise at different time points. (**A**) Percentage of performers for each breakfast (BF) style. The left-to-right columns are the Japanese, Japanese/Western, Western, cereal, and skipping breakfast groups. (**B**) Daily rhythm of performer percentage in Japanese BF (blue line) and other BF (orange line), including the J-W, western, and cereal groups. * *p* < 0.05, Fisher’s exact probability test vs. other BF groups. Irregular refers to exercise without specific time points.

**Table 1 foods-11-02609-t001:** Basic characteristics of participants.

		Male (Mean ± SD)	Female (Mean ± SD)	*p*
	Number	2352	1043	
	Age	49.7 ± 12	44.8 ± 12.7	<0.001
	BMI	23.3 ± 3.24	20.8 ± 3.09	<0.001
Exercise (METs)	Strong	473 ± 1114	288 ± 932	<0.001
	Moderate	295 ± 614	241 ± 608	<0.001
	Walk	780 ± 890	767 ± 945	NS
	Total	1548 ± 1948	1297 ± 1837	<0.001
Breakfast number (%)	Japanese	623 (26.4)	219 (21)	NS
	J-W	463 (19.7)	220 (21)	NS
	Western	860 (36.6)	392 (37.6)	NS
	Cereal	199 (8.5)	141 (13.5)	<0.001
	Skipping	208 (8.8)	75 (7.2)	NS
Sleep (clock time)	Weekday wake-up	5.99 ± 1.21	6.18 ± 1.13	<0.001
	Weekend wake-up	7.32 ± 1.67	7.78 ± 1.73	<0.001
	MSFsc	2.92 ± 1.57	3.20 ± 1.51	<0.001

All data are expressed as the mean ± SD, except for breakfast style, which showed percentage (%). Statistical significance was determined by Mann–Whitney analysis, except for the breakfast style, which was analyzed using Fisher’s exact probability test. NS; *p* > 0.05.

**Table 2 foods-11-02609-t002:** Spearman’s correlation analysis between basic characteristics, exercise, sleep parameters, and breakfast meal style.

	Japanese	J-W	Western	Cereal	Skipping
Age	−0.01	0.006	0.104 **	−0.073 **	−0.095 **
BMI	0.050 **	0.055 **	−0.043 *	−0.037 *	−0.044 *
Strong Exercise	0.075 **	0.015	−0.079 **	−0.009	0.009
Moderate Exercise	0.073 **	0.03	−0.066 **	−0.015	−0.027
Walk Exercise	−0.005	−0.011	0.019	0.02	−0.031
Total Exercise	0.036 *	−0.001	−0.041 *	0.031	−0.016
Weekday wake-up	−0.109 **	0.002	0.009	0.038 *	0.110 **
Weekday sleep onset	−0.124 **	−0.007	0.036 *	0.009	0.131 **
Weekday sleep length	0.054 **	−0.004	−0.033	0.026	−0.047 **
Weekday wake-up	−0.109 **	−0.011	0.003	0.049 *	0.128 **
Weekday sleep onset	−0.124 **	0.008	0.029	0.013	0.117 **
Weekday sleep length	0.017	−0.017	−0.042 *	0.037 *	0.031
MSFsc	−0.120 **	0.009	0.02	0.026	0.112 **

A higher score indicated a high association between the factors. Asterisks (* and **) in each column indicate the significance of the correlation (*p* < 0.05, *p* < 0.01). BMI, body mass index; MSFsc, a marker of morningness and eveningness; J-W, Japanese and Western meals.

**Table 3 foods-11-02609-t003:** Multivariate linear regression analysis for interaction between various exercise factors, sleep, and breakfast style.

Exercise	Breakfast	β	P	R2	F
Strong	Japanese	**0.067**	**0.025**	0.019	9.18
	J-W	0.048	0.090		
	Western	−0.005	0.885		
	Cereal	0.030	0.212		
Moderate	Japanese	**0.070**	**0.019**	0.005	3.12
	J-W	0.014	0.627		
	Western	−0.004	0.906		
	Cereal	0.000	0.994		
Walk	Japanese	0.032	0.286	0.008	4.27
	J-W	0.038	0.185		
	Western	0.031	0.329		
	Cereal	0.029	0.235		
Total	Japanese	**0.075**	**0.013**	0.008	4.57
	J-W	0.049	0.085		
	Western	0.011	0.728		
	Cereal	0.030	0.210		

For each subjective variable, the standardized coefficient (b) is indicated by a *p*-value. The R-squared and F-values were used to assess the fitness of the model. Confounding factors were age, sex, MSFsc, and BMI for physical activity observation, and those were age, sex, and BMI for MSFsc observation. J-W, Japanese and Western breakfast. Bold font in each column indicates the significance of this regression test (*p* < 0.05).

**Table 4 foods-11-02609-t004:** Binomial logistic regression analysis for the interaction between various exercise factors and breakfast style.

	Exercise			
	Strong	Moderate	Walk	Total
Breakfast	OR ^(a)^	OR ^(a)^	OR ^(a)^	OR ^(a)^
	95%CI ^(b)^	95%CI ^(b)^	95%CI ^(b)^	95%CI ^(b)^
Japanese	1.29	1.56 **	1.13	1.15
		1.18–2.05		
J-W	1.13	1.55 **	1.16	1.06
		1.17–2.06		
Western	0.85	1.08	1.28	0.94

Cereal	0.86	1.12	1.39 *	1.35
			1.01–1.91	

METs data in each strength was interpreted as binomial data as follows; (strong, moderate: METs = 0 or more; walk, total: METs < median or more). Confounding factors were age, sex, BMI, and MSFsc. (a) OR: Odds Ratio, (b) CI: Confidence Intervals. Asterisks (* and **) in each column indicate the significance of this regression test (*p* < 0.05, *p* < 0.01).

## Data Availability

Data will be sent upon request from the corresponding author. The data are not publicly available because of patent preparation.

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
