# Peer review of "Association between Breakfast Meal Categories and Timing of Physical Activity of Japanese Workers"

_foods, 2022, doi:10.3390/foods11172609_

Round 1

Reviewer 1 Report

The authors present a cross-sectional study of breakfast category relationship to physical activity among Japanese workers aged 20-69. This type of study is standard, although the categorisation of breakfast type is new to this reviewer. 

Minor points

P1 L25-26 The use of the reference "St-Onge et al. Meal Timing and Frequency: Implications for Cardiovascular Disease Prevention: A Scientific Statement From the American Heart Association. Circulation 2017, 135, e96–e121." is not supportive of the statement, given "In summary, the limited evidence of breakfast consumption as an important factor in combined weight and cardiometabolic risk management is suggestive of a minimal impact." The authors may choose to downgrade their assessment of the breakfast meal.

P1 L31-32 The authors imply that breakfast leads to better exercise performance e.g. "Therefore, eating breakfast has long-lasting effects and is particularly relevant for individuals who want to maximize their exercise performance". The associated citation does not mention breakfast.

The remaining introduction covers many points concisely and very well.

P2 L84-85 + P3 L96-98 Are there validation studies of the MCTQ (Munich Chronotype Questionnaire) and IPAQ with Japanese participants? Can these please be included and briefly discussed in the method? 

P4 Table 1. Spelling error in "moderate"

P4 L146-147 "Sleep factors showed strong negative associations with eating J (Table 2)". However the highest Spearmans association of J breakfast with sleep factors is -0.124, which is a very weak association, not strong.  

P7 L198-201 A statistical association of breakfast with physical activity or timing does not imply an importance.

P7 L201-206 Why mention dementia from the Thapa et al reference? The difference could be due to chronotype alone. 

P7 L216-217 "In addition, a study by Williams and Lamb indicated that breakfast may positively influence daily carbohydrate requirements" - What does "positively influence daily carbohydrate requirements" mean? Please be explicit as this statement is ambiguous. 

P7 L219-222 Some context is needed for this statement to assist the reader. The participants for this reference [45] were all male and trained runners - albeit amateur. But is an 9% increase in endurance capacity necessary for the current study population? Isn't time in MVPA sufficient?

P7 L229-231 Do these participants want to 'maximise' their performance? is it necessary? The questions have not been asked. 

P7 L234-236 The evidence does suggest, but not strongly.

P7 L244-245 Younger individuals also tend to have greater FFM:FM ratio than older individuals as a consequence of age muscle atrophy. 

Major points

P2 L70-71 There is an issue excluding 1331 participants "who did not have habitual physical activity". Reasoning for the exclusion was not provided. This is a significant proportion of the original participant group (n=5534). Physical activity (habitual or not) is a result and should be included. 

P3 L97-98 How was daily PA "objectively measured"? It appears that short form self-reported IPAQ was distributed (subjective) and not accelerometers (objective or similar). Self-report is not an objective measurement and does not normally match with objective measures such as accelerometry: Comparison of self-report and objective measures of physical activity in US adults with osteoarthritis - PubMed (nih.gov)

P4 Table 2 What is the correlation of physical activity with MSF(sc)? Importantly, Table 3 does not include MSF(sc) as a confounder, but in the discussion the authors state that morningness is associated with physical activity (P7 L232-234).  It is important that MSF(sc) is assessed for its association with physical activity and as a confounder.

P5 Fig 1.B What is the practical significance of the difference in MSF(sc)? I see that there is a statistical difference, however if statistical difference is small and has no practical significance, then does the difference matter? E.g. Males consuming J breakfast appeared to have a MSF(sc) ~0.7 difference with Skip and females MSF(sc) ~1.1 difference. What real-life scenario does this involve? How many hours difference in sleep or wake times does this difference manifest itself?

P7 L256-258 Could morningness be a predictor of J-type breakfast and higher MET's? The study: Development and initial validation of the Morningness-Eveningness Exercise Preference Questionnaire (MEEPQ) in Japanese university students | PLOS ONE certainly suggests that there is a relationship

Author Response

We would like to thank the editor and reviewers for reading our manuscript so carefully and providing valuable feedback. We improved the quality of our manuscript as a result of your courteous comments. Our responses are provided below in a point-by-point fashion and changes in the manuscripts are notated in red letters.

Reviewer 1

Query1

P1 L25-26 The use of the reference "St-Onge et al. Meal Timing and Frequency: Implications for Cardiovascular Disease Prevention: A Scientific Statement From the American Heart Association. Circulation 2017, 135, e96–e121." is not supportive of the statement, given "In summary, the limited evidence of breakfast consumption as an important factor in combined weight and cardiometabolic risk management is suggestive of a minimal impact." The authors may choose to downgrade their assessment of the breakfast meal.

Response1

Thank you for your valuable comment. We changed the sentences and rearranged them. Please see the lines 27-33.

Query2

P1 L31-32 The authors imply that breakfast leads to better exercise performance e.g. "Therefore, eating breakfast has long-lasting effects and is particularly relevant for individuals who want to maximize their exercise performance". The associated citation does not mention breakfast.

Response2

Thank you for your comments. We changed the order of the sentence to avoid misunderstanding to the reader. Please see lines 34-38.

Query3

P2 L84-85 + P3 L96-98 Are there validation studies of the MCTQ (Munich Chronotype Questionnaire) and IPAQ with Japanese participants? Can these please be included and briefly discussed in the method? 

Response3

 Thank you for your comments. We included the following papers in the methods. Please see the lines 89-92. In a study by Kitamura et al., the Japanese version of the MCTQ was developed to evaluate chronotype and the validity of the Japanese version of the MCTQ was established. (Kitamura, S., Hida, A., Aritake, S., Higuchi, S., Enomoto, M., Kato, M., Vetter, C., Roenneberg, T., & Mishima, K. (2014). Validity of the Japanese version of the Munich ChronoType Questionnaire. Chronobiology international, 31(7), 845–850. https://doi.org/10.3109/07420528.2014.914035)

Also, some studies concluded the validity of the International Physical Activity Questionnaire (IPAQ) for assessing physical activity in Japanese adults. We added these two reports below to the references. Please see the lines 119-120.

(Kimiko Tomioka, Junko Iwamoto, Keigo Saeki, Nozomi Okamoto, Reliability and Validity of the International Physical Activity Questionnaire (IPAQ) in Elderly Adults: The Fujiwara-kyo Study, Journal of Epidemiology, 2011, Volume 21, Issue 6, Pages 459-465, https://doi.org/10.2188/jea.JE20110003,

Murase, Norio & Katsumura, T. & Ueda, C. & Inoue, Shigeru & Shimomitsu, T.. (2002). International standardization of physical activity level: reliability and validity study of the Japanese version of the International Physical Activity Questionnaire (IPAQ) (Kosei no Shihyo). J.Health Welfare Stat.. 49.)

Query4

P4 Table 1. Spelling error in "moderate"

Response4

Thank you for your comment. We fixed the error.

Query5

P4 L146-147 "Sleep factors showed strong negative associations with eating J (Table 2)". However, the highest Spearmans association of J breakfast with sleep factors is -0.124, which is a very weak association, not strong.  

Response5

Thank you for your comment. We removed the word “strong”.

Query6

P7 L198-201 A statistical association of breakfast with physical activity or timing does not imply an importance.

Response6

We appreciate the reviewer’s comment and we changed the sentences. Please see lines 231-234.

Query7

P7 L201-206 Why mention dementia from the Thapa et al reference? The difference could be due to chronotype alone. 

Response7

We would like to thank  the reviewer’s valuable idea, and omitted the exceeded parts.

Query8

P7 L216-217 "In addition, a study by Williams and Lamb indicated that breakfast may positively influence daily carbohydrate requirements" - What does "positively influence daily carbohydrate requirements" mean? Please be explicit as this statement is ambiguous. 

Response8

We appreciate the reviewer’s comment and  changed the sentences and the references. Please see lines 238-239.

Query9

P7 L219-222 Some context is needed for this statement to assist the reader. The participants for this reference [45] were all male and trained runners - albeit amateur. But is an 9% increase in endurance capacity necessary for the current study population? Isn't time in MVPA sufficient?

Response9

Thank you for your important comments. We added some comments to assist the reader, and we changed the sentences. Please see lines 252-258.

Query10

P7 L229-231 Do these participants want to 'maximise' their performance? is it necessary? The questions have not been asked. 

Response10

Thank you for your valuable comment. One hypothesis of this study is that consuming breakfast would affect the performance or not. Therefore, maximizing the performance would be one of the objectives.

Query11

P7 L234-236 The evidence does suggest, but not strongly.

Response11

We appreciate the reviewer’s comment and removed the word “strongly”.

Query12

P7 L244-245 Younger individuals also tend to have greater FFM:FM ratio than older individuals as a consequence of age muscle atrophy. 

Response12

We appreciate the reviewer’s comment. We added this idea in the discussion. Please see the lines 281-283.

Major points

Query13

P2 L70-71 There is an issue excluding 1331 participants "who did not have habitual physical activity". Reasoning for the exclusion was not provided. This is a significant proportion of the original participant group (n=5534). Physical activity (habitual or not) is a result and should be included. 

Response13

We are sorry for misunderstanding this sentence. A total of 1331 participants did not write the answer to the question at IPAQ questionnaire, therefore we decided to exclude these participants’ data. Please see lines 73-74.

Query14

P3 L97-98 How was daily PA "objectively measured"? It appears that short form self-reported IPAQ was distributed (subjective) and not accelerometers (objective or similar). Self-report is not an objective measurement and does not normally match with objective measures such as accelerometry: Comparison of self-report and objective measures of physical activity in US adults with osteoarthritis - PubMed (nih.gov)

Response14

Thank you for your comment, we rewrote these sentences to avoid the misunderstanding. Please see lines 102-105.

Query15

P4 Table 2 What is the correlation of physical activity with MSF(sc)? Importantly, Table 3 does not include MSF(sc) as a confounder, but in the discussion the authors state that morningness is associated with physical activity (P7 L232-234).  It is important that MSF(sc) is assessed for its association with physical activity and as a confounder.

Response15

We appreciate the reviewer’s comment and conducted liner regression again including MSFsc as a confounder. Please see new Table 3.

Query16

P5 Fig 1.B What is the practical significance of the difference in MSF(sc)? I see that there is a statistical difference, however if statistical difference is small and has no practical significance, then does the difference matter? E.g. Males consuming J breakfast appeared to have a MSF(sc) ~0.7 difference with Skip and females MSF(sc) ~1.1 difference. What real-life scenario does this involve? How many hours difference in sleep or wake times does this difference manifest itself?

Response16

Thank you for your good comments. We added the data of sleep and wake time differences in Japanese and skip groups. We also referred to one paper which suggests the relation of 1 hr differences of MSFsc to muscle volume. Please see lines 234-241.

Query17

P7 L256-258 Could morningness be a predictor of J-type breakfast and higher MET's? The study: Development and initial validation of the Morningness-Eveningness Exercise Preference Questionnaire (MEEPQ) in Japanese university students | PLOS ONE certainly suggests that there is a relationship

Response17

We appreciate the reviewer’s comment. We added this comment in the discussion. Please see lines 297-305.

Reviewer 2 Report

This is a well-written manuscript. Below are my comments

Introduction:

I love how succint and to the point the introduction is. I appreciate the authors conciseness and their ability to provide the readers with a well-defined objective.

Methodology:

I appreciate the author's thoroughness in their methodology. There are just a couple of minor questions that I have regarding the methodology

1. Were participants compensated for their time? 

2. Were there questions built into the survey to determine whether participants were answering the questions truthfully?

3. Was the survey administered in English?

4. Can you please provide information about who received 276 vs 318 questions. A parenthetical reference will work

5. Did you apply any normalization techniques on non-normally distributed data?

6. Why did you not run a logistic regression? Based on what I am reading a logistic regression would have been your best choice.

Results

I appreciate the clearly written results however, I do have some issues with them.

Below are my comments

1. The sex and breakfast type correlation should be a tetrachloric correlation and not a Spearman since a Spearman does not make sense for a nominal variable

2. Based on the way the results are presented, a logistic linear regression would have been the most appropriate test for this study.

Discussion

I cannot judge the discussion because the statistical analyses were run incorrectly. 

Author Response

We would like to thank the editor and reviewers for reading our manuscript so carefully and providing valuable feedback. We improved the quality of our manuscript as a result of your courteous comments. Our responses are provided below in a point-by-point fashion and changes in the manuscripts are notated in red letters.

Reviewer 2

Reviewer’s comment

This is a well-written manuscript. Below are my comments

Reply

We appreciate the reviewer’s comments on our manuscript. Below, are the answerers to his questions.

Introduction

Reviewer’s comment

I love how succint and to the point the introduction is. I appreciate the authors conciseness and their ability to provide the readers with a well-defined objective.

Reply

Thank you for reading thoroughly and your favorable evaluation.

Methodology

Query1

Were participants compensated for their time? 

Response1

Thank you for your comments. We used an online survey company, and they paid some incentives to the participants.

Query2

Were there questions built into the survey to determine whether participants were answering the questions truthfully?

Response2

Thank you for your comments. We did not include such questions, but the survey company originally excluded the participants who did not complete the survey or answered the same number for all the questions.

Query3

Was the survey administered in English?

Response3

Thank you for your comments. The survey we used through this study was in Japanese. 

Query4

Can you please provide information about who received 276 vs 318 questions? A parenthetical reference will work

Response4

Thank you for your comments. The number of questions depended on the participant’s working condition (e.g., normal work, shift work, or remote homework). We added this information. Please see lines 83-86.

Query5

Did you apply any normalization techniques on non-normally distributed data?

Response5

Thank you for your comments. As you mentioned, we did normalization techniques on non-normally distributed data.

Query6

Why did you not run a logistic regression? Based on what I am reading a logistic regression would have been your best choice.

Response6

We appreciate the reviewer’s idea and conducted a logistic regression. Therefore, we added new Table 4.

Results

Reviewer’s comment

I appreciate the clearly written results however; I do have some issues with them.

Below are my comments

Reply

Thank you for reading thoroughly and your favorable evaluation. We could improve the quality thanks to your valuable comments. Please check the points we revised.

Query7

The sex and breakfast type correlation should be a tetrachloric correlation and not a Spearman since a Spearman does not make sense for a nominal variable.

Response7

We appreciate the reviewer’s comment, and we removed the sex in the correlation analysis. Please see new Table 2.

Query8

 Based on the way the results are presented, a logistic linear regression would have been the most appropriate test for this study. 

Response8

We appreciate the reviewer’s idea and conducted a binomial logistic regression, and we added this data as new Table 4.

Discussion

Reviewer’s comment

I cannot judge the discussion because the statistical analyses were run incorrectly. 

Reply

Thank you for your comment. We corrected the analysis and added new Tables, but we obtained similar results. We added some sentences in the discussion.

Round 2

Reviewer 2 Report

I appreciate the authors taking the time to address my concerns. Below are some concerns that need to be specifically address.

Please provide exactly how many questions were provided to each type of worker (i.e. shift work = xxx questions).

In the multi-variate logistic regressions did you account for age, sex and BMI?

Also why did you not include MSFsc in your analyses?  

Author Response

Reviewer 2

We would like to thank the editor and reviewer #2 for reading our manuscript so carefully and providing valuable feedback again. Our responses are provided below in a point-by-point fashion and changes in the manuscripts are notated in red letters.

Query1

Please provide exactly how many questions were provided to each type of worker (i.e. shift work = xxx questions).

Response1

We appreciate the reviewer’s comment and added detail explanation about number of the questions. Please see P2, L83-84.

Query2

In the multi-variate logistic regressions did you account for age, sex and BMI?

Response2

Thank you for your comment. We explained about cofounding factors you mentioned in P6, L184-186.

Query3

Also why did you not include MSFsc in your analyses? 

Response3

We appreciate the reviewer’s comment, and conducted multivariate linear regression analysis including MSFsc (please see table3). We added sentence about this observation. Please check the new Table.3 and see P5, L179.
